# Learning Robust Rewards with Adversarial Inverse Reinforcement Learning

**Justin Fu, Katie Luo, Sergey Levine**
Department of Electrical Engineering and Computer Science
University of California, Berkeley
Berkeley, CA 94720, USA
`justinjfu@eecs.berkeley.edu,katieluo@berkeley.edu,`
`svlevine@eecs.berkeley.edu`

## Abstract

Reinforcement learning provides a powerful and general framework for decision making and control, but its application in practice is often hindered by the need for extensive feature and reward engineering. Deep reinforcement learning methods can remove the need for explicit engineering of policy or value features, but still require a manually specified reward function. Inverse reinforcement learning holds the promise of automatic reward acquisition, but has proven exceptionally difficult to apply to large, high-dimensional problems with unknown dynamics. In this work, we propose AIRL, a practical and scalable inverse reinforcement learning algorithm based on an adversarial reward learning formulation. We demonstrate that AIRL is able to recover reward functions that are robust to changes in dynamics, enabling us to learn policies even under significant variation in the environment seen during training. Our experiments show that AIRL greatly outperforms prior methods in these transfer settings.

## 1 Introduction

While reinforcement learning (RL) provides a powerful framework for automating decision making and control, significant engineering of elements such as features and reward functions has typically been required for good practical performance. In recent years, deep reinforcement learning has alleviated the need for feature engineering for policies and value functions, and has shown promising results on a range of complex tasks, from vision-based robotic control (Levine et al., 2016) to video games such as Atari (Mnih et al., 2015) and Minecraft (Oh et al., 2016). However, reward engineering remains a significant barrier to applying reinforcement learning in practice. In some domains, this may be difficult to specify (for example, encouraging "socially acceptable" behavior), and in others, a naïvely specified reward function can produce unintended behavior (Amodei et al., 2016). Moreover, deep RL algorithms are often sensitive to factors such as reward sparsity and magnitude, making well performing reward functions particularly difficult to engineer.

Inverse reinforcement learning (IRL) (Russell, 1998; Ng & Russell, 2000) refers to the problem of inferring an expert's reward function from demonstrations, which is a potential method for solving the problem of reward engineering. However, inverse reinforcement learning methods have generally been less efficient than direct methods for learning from demonstration such as imitation learning (Ho & Ermon, 2016), and methods using powerful function approximators such as neural networks have required tricks such as domain-specific regularization and operate inefficiently over whole trajectories (Finn et al., 2016b). There are many scenarios where IRL may be preferred over direct imitation learning, such as re-optimizing a reward in novel environments (Finn et al., 2017) or to infer an agent's intentions, but IRL methods have not been shown to scale to the same complexity of tasks as direct imitation learning. However, adversarial IRL methods (Finn et al., 2016b;a) hold promise for tackling difficult tasks due to the ability to adapt training samples to improve learning efficiency.

Part of the challenge is that IRL is an ill-defined problem, since there are many optimal policies that can explain a set of demonstrations, and many rewards that can explain an optimal policy (Ng

et al., 1999). The maximum entropy (MaxEnt) IRL framework introduced by Ziebart et al. (2008) handles the former ambiguity, but the latter ambiguity means that IRL algorithms have difficulty distinguishing the true reward functions from those shaped by the environment dynamics. While shaped rewards can increase learning speed in the original training environment, when the reward is deployed at test-time on environments with varying dynamics, it may no longer produce optimal behavior, as we discuss in Sec. 5. To address this issue, we discuss how to modify IRL algorithms to learn rewards that are invariant to changing dynamics, which we refer to as *disentangled rewards*.

In this paper, we propose adversarial inverse reinforcement learning (AIRL), an inverse reinforcement learning algorithm based on adversarial learning. Our algorithm provides for simultaneous learning of the reward function and value function, which enables us to both make use of the efficient adversarial formulation and recover a generalizable and portable reward function, in contrast to prior works that either do not recover a reward functions (Ho & Ermon, 2016), or operates at the level of entire trajectories, making it difficult to apply to more complex problem settings (Finn et al., 2016b;a). Our experimental evaluation demonstrates that AIRL outperforms prior IRL methods (Finn et al., 2016b) on continuous, high-dimensional tasks with unknown dynamics by a wide margin. When compared to GAIL (Ho & Ermon, 2016), which does not attempt to directly recover rewards, our method achieves comparable results on tasks that do not require transfer. However, on tasks where there is considerable variability in the environment from the demonstration setting, GAIL and other IRL methods fail to generalize. In these settings, our approach, which can effectively disentangle the goals of the expert from the dynamics of the environment, achieves superior results.

## 2 RELATED WORK

Inverse reinforcement learning (IRL) is a form of imitation learning and learning from demonstration (Argall et al., 2009). Imitation learning methods seek to learn policies from expert demonstrations, and IRL methods accomplish this by first inferring the expert's reward function. Previous IRL approaches have included maximum margin approaches (Abbeel & Ng, 2004; Ratliff et al., 2006), and probabilistic approaches such as Ziebart et al. (2008); Boularias et al. (2011). In this work, we work under the maximum causal IRL framework of Ziebart (2010). Some advantages of this framework are that it removes ambiguity between demonstrations and the expert policy, and allows us to cast the reward learning problem as a maximum likelihood problem, connecting IRL to generative model training.

Our proposed method most closely resembles the algorithms proposed by Uchibe (2017); Ho & Ermon (2016); Finn et al. (2016a). Generative adversarial imitation learning (GAIL) (Ho & Ermon, 2016) differs from our work in that it is not an IRL algorithm that seeks to recover reward functions. The critic or discriminator of GAIL is unsuitable as a reward since, at optimality, it outputs 0.5 uniformly across all states and actions. Instead, GAIL aims only to recover the expert's policy, which is a less portable representation for transfer. Uchibe (2017) does not interleave policy optimization with reward learning within an adversarial framework. Improving a policy within an adversarial framework corresponds to training an amortized sampler for an energy-based model, and prior work has shown this is crucial for performance (Finn et al., 2016b). Wulfmeier et al. (2015) also consider learning cost functions with neural networks, but only evaluate on simple domains where analytically solving the problem with value iteration is tractable. Previous methods which aim to learn nonlinear cost functions have used boosting (Ratliff et al., 2007) and Gaussian processes (Levine et al., 2011), but still suffer from the feature engineering problem.

Our IRL algorithm builds on the adversarial IRL framework proposed by Finn et al. (2016a), with the discriminator corresponding to an odds ratio between the policy and exponentiated reward distribution. The discussion in Finn et al. (2016a) is theoretical, and to our knowledge no prior work has reported a practical implementation of this method. Our experiments show that direct implementation of the proposed algorithm is ineffective, due to high variance from operating over entire trajectories. While it is straightforward to extend the algorithm to single state-action pairs, as we discuss in Section 4, a simple unrestricted form of the discriminator is susceptible to the reward ambiguity described in (Ng et al., 1999), making learning the portable reward functions difficult. As illustrated in our experiments, this greatly limits the generalization capability of the method: the learned reward functions are not robust to environment changes, and it is difficult to use the algo-

rithm for the purpose of inferring the intentions of agents. We discuss how to overcome this issue in Section 5.

Amin et al. (2017) consider learning reward functions which generalize to new tasks given multiple training tasks. Our work instead focuses on how to achieve generalization within the standard IRL formulation.

## 3 BACKGROUND

Our inverse reinforcement learning method builds on the maximum causal entropy IRL framework (Ziebart, 2010), which considers an entropy-regularized Markov decision process (MDP), defined by the tuple $(\mathcal{S}, \mathcal{A}, \mathcal{T}, r, \gamma, \rho_0)$. $\mathcal{S}, \mathcal{A}$ are the state and action spaces, respectively, $\gamma \in (0, 1)$ is the discount factor. The dynamics or transition distribution $\mathcal{T}(s'|a, s)$, the initial state distribution $\rho_0(s)$, and the reward function $r(s, a)$ are unknown in the standard reinforcement learning setup and can only be queried through interaction with the MDP.

The goal of (forward) reinforcement learning is to find the optimal policy $\pi^*$ that maximizes the expected entropy-regularized discounted reward, under $\pi$, $\mathcal{T}$, and $\rho_0$:

$$\pi^* = \arg\max_\pi E_{\tau \sim \pi} \left[ \sum_{t=0}^T \gamma^t (r(s_t, a_t) + H(\pi(\cdot|s_t))) \right],$$

where $\tau = (s_0, a_0, ...s_T, a_T)$ denotes a sequence of states and actions induced by the policy and dynamics. It can be shown that the trajectory distribution induced by the optimal policy $\pi^*(a|s)$ takes the form $\pi^*(a|s) \propto \exp\{Q^*_{\text{soft}}(s_t, a_t)\}$ (Ziebart, 2010; Haarnoja et al., 2017), where $Q^*_{\text{soft}}(s_t, a_t) = r_t(s, a) + E_{(s_{t+1},...)\sim\pi}[\sum_{t'=t}^T \gamma^{t'} (r(s_{t'}, a_{t'}) + H(\pi(\cdot|s_{t'})))]$ denotes the soft Q-function.

Inverse reinforcement learning instead seeks infer the reward function $r(s, a)$ given a set of demonstrations $\mathcal{D} = \{\tau_1, ..., \tau_N\}$. In IRL, we assume the demonstrations are drawn from an optimal policy $\pi^*(a|s)$. We can interpret the IRL problem as solving the maximum likelihood problem:

$$\max_\theta E_{\tau \sim \mathcal{D}} \left[ \log p_\theta(\tau) \right] , \tag{1}$$

Where $p_\theta(\tau) \propto p(s_0) \prod_{t=0}^T p(s_{t+1}|s_t, a_t) e^{\gamma^t r_\theta(s_t, a_t)}$ parametrizes the reward function $r_\theta(s, a)$ but fixes the dynamics and initial state distribution to that of the MDP. Note that under deterministic dynamics, this simplifies to an energy-based model where for feasible trajectories, $p_\theta(\tau) \propto e^{\sum_{t=0}^T \gamma^t r_\theta(s_t, a_t)}$ (Ziebart et al., 2008).

Finn et al. (2016a) propose to cast optimization of Eqn. 1 as a GAN (Goodfellow et al., 2014) optimization problem. They operate in a trajectory-centric formulation, where the discriminator takes on a particular form ($f_\theta(\tau)$ is a learned function; $\pi(\tau)$ is precomputed and its value "filled in"):

$$D_\theta(\tau) = \frac{\exp\{f_\theta(\tau)\}}{\exp\{f_\theta(\tau)\} + \pi(\tau)}, \tag{2}$$

and the policy $\pi$ is trained to maximize $R(\tau) = \log(1 - D(\tau)) - \log D(\tau)$. Updating the discriminator can be viewed as updating the reward function, and updating the policy can be viewed as improving the sampling distribution used to estimate the partition function. If trained to optimality, it can be shown that an optimal reward function can be extracted from the optimal discriminator as $f^*(\tau) = R^*(\tau) + \text{const}$, and $\pi$ recovers the optimal policy. We refer to this formulation as generative adversarial network guided cost learning (GAN-GCL) to discriminate it from guided cost learning (GCL) (Finn et al., 2016a). This formulation shares similarities with GAIL (Ho & Ermon, 2016), but GAIL does not place special structure on the discriminator, so the reward cannot be recovered.

## 4 ADVERSARIAL INVERSE REINFORCEMENT LEARNING (AIRL)

In practice, using full trajectories as proposed by GAN-GCL can result in high variance estimates as compared to using single state, action pairs, and our experimental results show that this results in

very poor learning. We could instead propose a straightforward conversion of Eqn. 2 into the single state and action case, where:

$$D_\theta(s, a) = \frac{\exp\{f_\theta(s, a)\}}{\exp\{f_\theta(s, a)\} + \pi(a|s)}.$$

As in the trajectory-centric case, we can show that, at optimality, $f^*(s, a) = \log \pi^*(a|s) = A^*(s, a)$, the advantage function of the optimal policy. We justify this, as well as a proof that this algorithm solves the IRL problem in Appendix A .

This change results in an efficient algorithm for imitation learning. However, it is less desirable for the purpose of reward learning. While the advantage is a valid optimal reward function, it is a heavily *entangled* reward, as it supervises each action based on the action of the optimal policy for the training MDP. Based on the analysis in the following Sec. 5, we cannot guarantee that this reward will be robust to changes in environment dynamics. In our experiments we demonstrate several cases where this reward simply encourages mimicking the expert policy $\pi^*$, and fails to produce desirable behavior even when changes to the environment are made.

## 5 THE REWARD AMBIGUITY PROBLEM

We now discuss why IRL methods can fail to learn robust reward functions. First, we review the concept of reward shaping. Ng et al. (1999) describe a class of reward transformations that preserve the optimal policy. Their main theoretical result is that under the following reward transformation,

$$\hat{r}(s, a, s') = r(s, a, s') + \gamma\Phi(s') - \Phi(s) , \tag{3}$$

the optimal policy remains unchanged, for any function $\Phi : \mathcal{S} \to \mathbb{R}$. Moreover, without prior knowledge of the dynamics, this is the *only* class of reward transformations that exhibits policy invariance. Because IRL methods only infer rewards from demonstrations given from an optimal agent, they cannot in general disambiguate between reward functions within this class of transformations, unless the class of learnable reward functions is restricted.

We argue that shaped reward functions may not be robust to changes in dynamics. We formalize this notion by studying policy invariance in two MDPs $M, M'$ which share the same reward and differ only in the dynamics, denoted as $T$ and $T'$, respectively.

Suppose an IRL algorithm recovers a shaped, policy invariant reward $\hat{r}(s, a, s')$ under MDP $M$ where $\Phi \neq 0$. Then, there exists MDP pairs $M, M'$ where changing the transition model from $T$ to $T'$ breaks policy invariance on MDP $M'$. As a simple example, consider deterministic dynamics $T(s, a) \to s'$ and state-action rewards $\hat{r}(s, a) = r(s, a) + \gamma\Phi(T(s, a)) - \Phi(s)$. It is easy to see that changing the dynamics $T$ to $T'$ such that $T'(s, a) \neq T(s, a)$ means that $\hat{r}(s, a)$ no longer lies in the equivalence class of Eqn. 3 for $M'$.

### 5.1 DISENTANGLING REWARDS FROM DYNAMICS

First, let the notation $Q^*_{r,T}(s, a)$ denote the optimal Q-function with respect to a reward function $r$ and dynamics $T$, and $\pi^*_{r,T}(a|s)$ denote the same for policies. We first define our notion of a "disentangled" reward.

**Definition 5.1** (Disentangled Rewards). *A reward function $r'(s, a, s')$ is (perfectly) disentangled with respect to a ground-truth reward $r(s, a, s')$ and a set of dynamics $\mathcal{T}$ such that under all dynamics $T \in \mathcal{T}$, the optimal policy is the same: $\pi^*_{r',T}(a|s) = \pi^*_{r,T}(a|s)$*

We could also expand this definition to include a notion of suboptimality. However, we leave this direction to future work.

Under maximum causal entropy RL, the following condition is equivalent to two optimal policies being equal, since Q-functions and policies are equivalent representations (up to arbitrary functions of state $f(s)$):

$$Q^*_{r',T}(s, a) = Q^*_{r,T}(s, a) - f(s)$$

To remove unwanted reward shaping with arbitrary reward function classes, the learned reward function can only depend on the current state $s$. We require that the dynamics satisfy a decomposability

---

**Algorithm 1** Adversarial inverse reinforcement learning

1: Obtain expert trajectories $\tau_i^E$
2: Initialize policy $\pi$ and discriminator $D_{\theta,\phi}$.
3: **for** step $t$ in $\{1, \ldots, N\}$ **do**
4:     Collect trajectories $\tau_i = (s_0, a_0, ..., s_T, a_T)$ by executing $\pi$.
5:     Train $D_{\theta,\phi}$ via binary logistic regression to classify expert data $\tau_i^E$ from samples $\tau_i$.
6:     Update reward $r_{\theta,\phi}(s, a, s') \leftarrow \log D_{\theta,\phi}(s, a, s') - \log(1 - D_{\theta,\phi}(s, a, s'))$
7:     Update $\pi$ with respect to $r_{\theta,\phi}$ using any policy optimization method.
8: **end for**

---

condition where functions over current states $f(s)$ and next states $g(s')$ can be isolated from their sum $f(s) + g(s')$. This can be satisfied for example by adding self transitions at each state to an ergodic MDP, or any of the environments used in our experiments. The exact definition of the condition, as well as proof of the following statements are included in Appendix B.

**Theorem 5.1.** *Let $r(s)$ be a ground-truth reward, and $T$ be a dynamics model satisfying the decomposability condition. Suppose IRL recovers a state-only reward $r'(s)$ such that it produces an optimal policy in $T$:*

$$Q_{r',T}^*(s, a) = Q_{r,T}^*(s, a) - f(s)$$

*Then, $r'(s)$ is disentangled with respect to all dynamics.*

**Theorem 5.2.** *If a reward function $r'(s, a, s')$ is disentangled for all dynamics functions, then it must be state-only. i.e. If for all dynamics $T$,*

$$Q_{r,T}^*(s, a) = Q_{r',T}^*(s, a) + f(s) \; \forall s, a$$

*Then $r'$ is only a function of state.*

In the traditional IRL setup, where we learn the reward in a single MDP, our analysis motivates learning reward functions that are solely functions of state. If the ground truth reward is also only a function of state, this allows us to recover the true reward up to a constant.

## 6   LEARNING DISENTANGLED REWARDS WITH AIRL

In the method presented in Section 4, we cannot learn a state-only reward function, $r_\theta(s)$, meaning that we cannot guarantee that learned rewards will not be shaped. In order to decouple the reward function from the advantage, we propose to modify the discriminator of Sec. 4 with the form:

$$D_{\theta,\phi}(s, a, s') = \frac{\exp\{f_{\theta,\phi}(s, a, s')\}}{\exp\{f_{\theta,\phi}(s, a, s')\} + \pi(a|s)},$$

where $f_{\theta,\phi}$ is restricted to a reward approximator $g_\theta$ and a shaping term $h_\phi$ as

$$f_{\theta,\phi}(s, a, s') = g_\theta(s, a) + \gamma h_\phi(s') - h_\phi(s). \tag{4}$$

The additional shaping term helps mitigate the effects of unwanted shaping on our reward approximator $g_\theta$ (and as we will show, in some cases it can account for *all* shaping effects). The entire training procedure is detailed in Algorithm 1. Our algorithm resembles GAIL (Ho & Ermon, 2016) and GAN-GCL (Finn et al., 2016a), where we alternate between training a discriminator to classify expert data from policy samples, and update the policy to confuse the discriminator.

The advantage of this approach is that we can now parametrize $g_\theta(s)$ as solely a function of the state, allowing us to extract rewards that are disentangled from the dynamics of the environment in which they were trained. In fact, under this restricted case, we can show the following under deterministic environments with a state-only ground truth reward (proof in Appendix C):

$$g^*(s) = r^*(s) + \text{const},$$

$$h^*(s) = V^*(s) + \text{const},$$

where $r^*$ is the *true* reward function. Since $f^*$ must recover to the advantage as shown in Sec. 4, $h$ recovers the optimal value function $V^*$, which serves as the reward shaping term.

To be consistent with Sec. 4, an alternative way to interpret the form of Eqn. 4 is to view $f_{\theta,\phi}$ as the advantage under deterministic dynamics

$$f^*(s, a, s') = \underbrace{r^*(s) + \gamma V^*(s')}_{Q(s,a)} - \underbrace{V^*(s)}_{V(s)} = A^*(s, a)$$

In stochastic environments, we can instead view $f(s, a, s')$ as a single-sample estimate of $A^*(s, a)$.

## 7 EXPERIMENTS

In our experiments, we aim to answer two questions:

1. Can AIRL learn disentangled rewards that are robust to changes in environment dynamics?

2. Is AIRL efficient and scalable to high-dimensional continuous control tasks?

To answer 1, we evaluate AIRL in transfer learning scenarios, where a reward is learned in a training environment, and optimized in a test environment with significantly different dynamics. We show that rewards learned with our algorithm under the constraint presented in Section 5 still produce optimal or near-optimal behavior, while naïve methods that do not consider reward shaping fail. We also show that in small MDPs, we can recover the exact ground truth reward function.

To answer 2, we compare AIRL as an imitation learning algorithm against GAIL (Ho & Ermon, 2016) and the GAN-based GCL algorithm proposed by Finn et al. (2016a), which we refer to as GAN-GCL, on standard benchmark tasks that do not evaluate transfer. Note that Finn et al. (2016a) does not implement or evaluate GAN-GCL and, to our knowledge, we present the first empirical evaluation of this algorithm. We find that AIRL performs on par with GAIL in a traditional imitation learning setup while vastly outperforming it in transfer learning setups, and outperforms GAN-GCL in both settings. It is worth noting that, except for (Finn et al., 2016b), our method is the only IRL algorithm that we are aware of that scales to high dimensional tasks with unknown dynamics, and although GAIL (Ho & Ermon, 2016) resembles an IRL algorithm in structure, it does not recover disentangled reward functions, making it unable to re-optimize the learned reward under changes in the environment, as we illustrate below.

For our continuous control tasks, we use trust region policy optimization (Schulman et al., 2015) as our policy optimization algorithm across all evaluated methods, and in the tabular MDP task, we use soft value iteration. We obtain expert demonstrations by training an expert policy on the ground truth reward, but hide the ground truth reward from the IRL algorithm. In this way, we simulate a scenario where we wish to use RL to solve a task but wish to refrain from manual reward engineering and instead seek to learn a reward function from demonstrations. Our code and additional supplementary material including videos will be available at `https://sites.google.com/view/adversarial-irl`, and hyper-parameter and architecture choices are detailed in Appendix D.

### 7.1 RECOVERING TRUE REWARDS IN TABULAR MDPs

We first consider MaxEnt IRL in a toy task with randomly generated MDPs. The MDPs have 16 states, 4 actions, randomly drawn transition matrices, and a reward function that always gives a reward of 1.0 when taking an action from state 0. The initial state is always state 1.

The optimal reward, learned reward with a state-only reward function, and learned reward using a state-action reward function are shown in Fig. 1. We subtract a constant offset from all reward functions so that they share the same mean for visualization - this does not influence the optimal policy. AIRL with a state-only reward function is able to recover the ground truth reward, but AIRL with a state-action reward instead recovers a shaped advantage function.

We also show that in the transfer learning setup, under a new transition matrix $T'$, the optimal policy under the state-only reward achieves optimal performance (it is identical to the ground truth reward) whereas the state-action reward only improves marginally over uniform random policy. The learning curve for this experiment is shown in Fig 2.

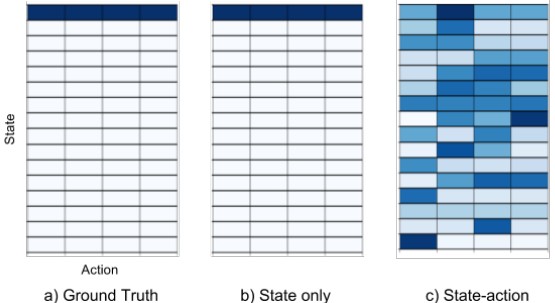

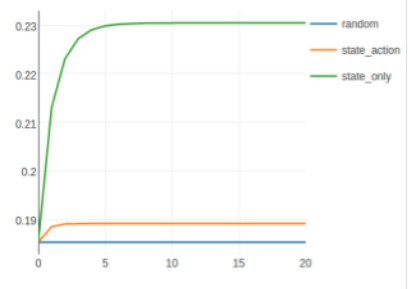

Figure 1: Ground truth (a) and learned rewards (b, c) on the random MDP task. Dark blue corresponds to a reward of 1, and white corresponds to 0. Note that AIRL with a state-only reward recovers the ground truth, whereas the state-action reward is shaped.

Figure 2: Learning curve for the transfer learning experiment on tabular MDPs. Value iteration steps are plotted on the x-axis, against returns for the policy on the y-axis.

## 7.2 DISENTANGLING REWARDS IN CONTINUOUS CONTROL TASKS

To evaluate whether our method can learn disentangled rewards in higher dimensional environments, we perform transfer learning experiments on continuous control tasks. In each task, a reward is learned via IRL on the training environment, and the reward is used to reoptimize a new policy on a test environment. We train two IRL algorithms, AIRL and GAN-GCL, with state-only and state-action rewards. We also include results for directly transferring the policy learned with GAIL, and an oracle result that involves optimizing the ground truth reward function with TRPO. Numerical results for these environment transfer experiments are given in Table 1.

The first task involves a 2D point mass navigating to a goal position in a small maze when the position of the walls are changed between train and test time. At test time, the agent cannot simply mimic the actions learned during training, and instead must successfully infer that the goal in the maze is to reach the target. The task is shown in Fig. 3. Only AIRL trained with state-only rewards is able to consistently navigate to the goal when the maze is modified. Direct policy transfer and state-action IRL methods learn rewards which encourage the agent to take the same path taken in the training environment, which is blocked in the test environment. We plot the learned reward in Fig. 4.

In our second task, we modify the agent itself. We train a quadrupedal "ant" agent to run forwards, and at test time we disable and shrink two of the front legs of the ant such that it must significantly change its gait.We find that AIRL is able to learn reward functions that encourage the ant to move forwards, acquiring a modified gait that involves orienting itself to face the forward direction and crawling with its two hind legs. Alternative methods, including transferring a policy learned by GAIL (which achieves near-optimal performance with the unmodified agent), fail to move forward at all. We show the qualitative difference in behavior in Fig. 5.

We have demonstrated that AIRL can learn disentangled rewards that can accommodate significant domain shift even in high-dimensional environments where it is difficult to exactly extract the true reward. GAN-GCL can presumably learn disentangled rewards, but we find that the trajectory-centric formulation does not perform well even in learning rewards in the original task, let alone transferring to a new domain. GAIL learns successfully in the training domain, but does not acquire a representation that is suitable for transfer to test domains.

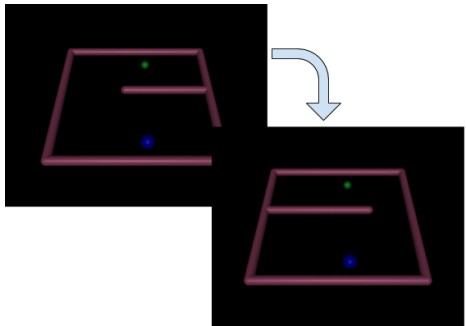

Figure 3: Illustration of the shifting maze task, where the agent (blue) must reach the goal (green). During training the agent must go around the wall on the left side, but during test time it must go around on the right.

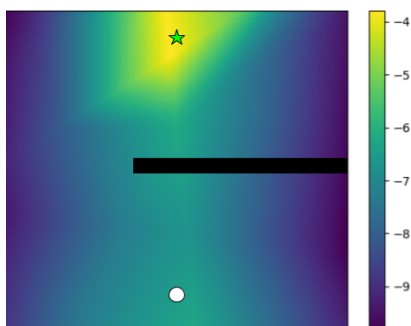

Figure 4: Reward learned on the point mass shifting maze task. The goal is located at the green star and the agent starts at the white circle. Note that there is little reward shaping, which enables the reward to transfer well.

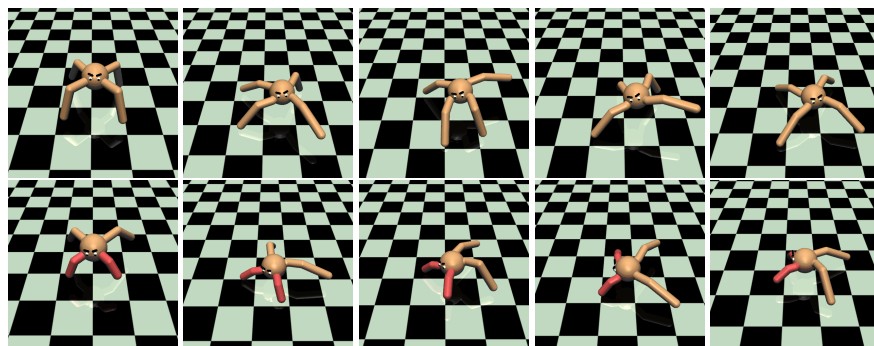

Figure 5: *Top row*: An ant running forwards (right in the picture) in the training environment. *Bottom row*: Behavior acquired by optimizing a state-only reward learned with AIRL on the disabled ant environment. Note that the ant must orient itself before crawling forward, which is a qualitatively different behavior from the optimal policy in the original environment, which runs sideways.

Table 1: Results on transfer learning tasks. Mean scores (higher is better) are reported over 5 runs. We also include results for TRPO optimizing the ground truth reward, and the performance of a policy learned via GAIL on the training environment.

|  | State-Only? | Point Mass-Maze | Ant-Disabled |
|---|---|---|---|
| GAN-GCL | No | -40.2 | -44.8 |
| GAN-GCL | Yes | -41.8 | -43.4 |
| AIRL (**ours**) | No | -31.2 | -41.4 |
| AIRL (**ours**) | Yes | **-8.82** | **130.3** |
| GAIL, policy transfer | N/A | -29.9 | -58.8 |
| TRPO, ground truth | N/A | -8.45 | 315.5 |

## 7.3 BENCHMARK TASKS FOR IMITATION LEARNING

Finally, we evaluate AIRL as an imitation learning algorithm against the GAN-GCL and the state-of-the-art GAIL on several benchmark tasks. Each algorithm is presented with 50 expert demonstrations, collected from a policy trained with TRPO on the ground truth reward function. For AIRL, we use an unrestricted state-action reward function as we are not concerned with reward transfer. Numerical results are presented in Table 2.These experiments do not test transfer, and in a sense can be regarded as "testing on the training set," but they match the settings reported in prior work (Ho & Ermon, 2016).

We find that the performance difference between AIRL and GAIL is negligible, even though AIRL is a true IRL algorithm that recovers reward functions, while GAIL does not. Both methods achieve close to the best possible result on each task, and there is little room for improvement. This result goes against the belief that IRL algorithms are indirect, and less efficient that direct imitation learning algorithms (Ho & Ermon, 2016). The GAN-GCL method is ineffective on all but the simplest Pendulum task when trained with the same number of samples as AIRL and GAIL. We find that a discriminator trained over trajectories easily overfits and provides poor learning signal for the policy.

Our results illustrate that AIRL achieves the same performance as GAIL on benchmark imitation tasks that do not require any generalization. On tasks that require transfer and generalization, illustrated in the previous section, AIRL outperforms GAIL by a wide margin, since our method is able to recover disentangled rewards that transfer effectively in the presence of domain shift.

Table 2: Results on imitation learning benchmark tasks. Mean scores (higher is better) are reported across 5 runs.

|                      | Pendulum   | Ant        | Swimmer    | Half-Cheetah |
|----------------------|------------|------------|------------|--------------|
| GAN-GCL              | -261.5     | 460.6      | -10.6      | -670.7       |
| GAIL                 | -226.0     | **1358.7** | **140.2**  | 1642.8       |
| AIRL (**ours**)      | **-204.7** | 1238.6     | 139.1      | **1839.8**   |
| AIRL State Only (**ours**) | -221.5 | 1089.3   | 136.4      | 891.9        |
| Expert (TRPO)        | -179.6     | 1537.9     | 141.1      | 1811.2       |
| Random               | -654.5     | -108.1     | -11.5      | -988.4       |

## 8   CONCLUSION

We presented AIRL, a practical and scalable IRL algorithm that can learn disentangled rewards and greatly outperforms both prior imitation learning and IRL algorithms. We show that rewards learned with AIRL transfer effectively under variation in the underlying domain, in contrast to unmodified IRL methods which tend to recover brittle rewards that do not generalize well and GAIL, which does not recover reward functions at all. In small MDPs where the optimal policy and reward are unambiguous, we also show that we can exactly recover the ground-truth rewards up to a constant.

## ACKNOWLEDGEMENTS

This research was supported by the National Science Foundation through IIS-1651843, IIS-1614653, and IIS-1637443. We would like to thank Roberto Calandra for helpful feedback on the paper.

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

## APPENDICES

## A   JUSTIFICATION OF AIRL

In this section, we show that the objective of AIRL matches that of solving the maximum causal entropy IRL problem. We use a similar method as Finn et al. (2016a), which shows the justification of GAN-GCL for the trajectory-centric formulation. For simplicity we derive everything in the undiscounted case.

### A.1   SETUP

As mentioned in Section 3, the goal of IRL can be seen as training a generative model over trajectories as:

$$\max_\theta J(\theta) = \max_\theta E_{\tau \sim \mathcal{D}}[\log p_\theta(\tau)]$$

Where the distribution $p_\theta(\tau)$ is parametrized as $p_\theta(\tau) \propto p(s_0) \prod_{t=0}^{T-1} p(s_{t+1}|s_t, a_t) e^{r_\theta(s_t, a_t)}$. We can compute the gradient with respect to $\theta$ as follows:

$$\frac{\partial}{\partial \theta} J(\theta) = E_{\mathcal{D}}[\frac{\partial}{\partial \theta} \log p_\theta(\tau)] == E_{\mathcal{D}}[\sum_{t=0}^{T} \frac{\partial}{\partial \theta} r_\theta(s_t, a_t)] - \frac{\partial}{\partial \theta} \log Z_\theta$$

$$= E_{\mathcal{D}}[\sum_{t=0}^{T} \frac{\partial}{\partial \theta} r_\theta(s_t, a_t)] - E_{p_\theta}[\sum_{t=0}^{T} \frac{\partial}{\partial \theta} r_\theta(s_t, a_t)]$$

Let $p_{\theta,t}(s_t, a_t) = \int_{s_{t' \neq t}, a_{t' \neq t}} p_\theta(\tau)$ denote the state-action marginal at time $t$. Rewriting the above equation, we have:

$$\frac{\partial}{\partial \theta} J(\theta) = \sum_{t=0}^{T} E_{\mathcal{D}}[\frac{\partial}{\partial \theta} r_\theta(s_t, a_t)] - E_{p_{\theta,t}}[\frac{\partial}{\partial \theta} r_\theta(s_t, a_t)]$$

As it is difficult to draw samples from $p_\theta$, we instead train a separate importance sampling distribution $\mu(\tau)$. For the choice of this distribution, we follow Finn et al. (2016a) and use a mixture policy $\mu(a|s) = \frac{1}{2}\pi(a|s) + \frac{1}{2}\hat{p}(a|s)$, where $\hat{p}(a|s)$ is a rough density estimate trained on the demonstrations. This is justified as reducing the variance of the importance sampling estimate when the policy $\pi(a|s)$ has poor coverage over the demonstrations in the early stages of training. Thus, our new gradient is:

$$\frac{\partial}{\partial \theta} J(\theta) = \sum_{t=0}^{T} E_{\mathcal{D}}[\frac{\partial}{\partial \theta} r_\theta(s_t, a_t)] - E_{\mu_t}[\frac{p_{\theta,t}(s_t, a_t)}{\mu_t(s_t, a_t)} \frac{\partial}{\partial \theta} r_\theta(s_t, a_t)] \tag{5}$$

We additionally wish to adapt the importance sampler $\pi$ to reduce variance, by minimizing $D_{KL}(\pi(\tau)||p_\theta(\tau))$. The policy trajectory distribution factorizes as $\pi(\tau) = p(s_0) \prod_{t=0}^{T-1} p(s_{t+1}|s_t, a_t)\pi(a_t|s_t)$. The dynamics and initial state terms inside $\pi(\tau)$ and $p_\theta(\tau)$ cancel, leaving the entropy-regularized policy objective:

$$\max_\pi E_\pi[\sum_{t=0}^{T} r_\theta(s_t, a_t) - \log \pi(a_t|s_t))] \tag{6}$$

In AIRL, we replace the cost learning objective with training a discriminator of the following form:

$$D_\theta(s, a) = \frac{\exp\{f_\theta(s, a)\}}{\exp\{f_\theta(s, a)\} + \pi(a|s)} \tag{7}$$

The objective of the discriminator is to minimize cross-entropy loss between expert demonstrations and generated samples:

$$\mathcal{L}(\theta) = \sum_{t=0}^{T} -E_{\mathcal{D}}[\log D_\theta(s_t, a_t)] - E_{\pi_t}[\log(1 - D_\theta(s_t, a_t))]$$

We replace the policy optimization objective with the following reward:

$$\hat{r}(s, a) = \log(D_\theta(s, a)) - \log(1 - D_\theta(s, a))$$

## A.2 DISCRIMINATOR OBJECTIVE

First, we show that training the gradient of the discriminator objective is the same as Eqn. 5. We write the negative loss to turn the minimization problem into maximization, and use $\mu$ to denote a mixture between the dataset and policy samples.

$$
\begin{aligned}
-\mathcal{L}(\theta) &= \sum_{t=0}^{T} E_\mathcal{D}\left[\log D_\theta(s_t, a_t)\right] + E_{\pi_t}\left[\log(1 - D_\theta(s_t, a_t))\right] \\
&= \sum_{t=0}^{T} E_\mathcal{D}\left[\log \frac{\exp\{f_\theta(s_t, a_t)\}}{\exp\{f_\theta(s_t, a_t)\} + \pi(a_t|s_t)}\right] + E_{\pi_t}\left[\log \frac{\pi(a_t|s_t)}{\exp\{f_\theta(s_t, a_t)\} + \pi(a_t|s_t)}\right] \\
&= \sum_{t=0}^{T} E_\mathcal{D}\left[f_\theta(s_t, a_t)\right] + E_{\pi_t}\left[\log \pi(a_t|s_t)\right] - 2E_{\bar{\mu}_t}\left[\log\left(\exp\{f_\theta(s_t, a_t)\} + \pi(a_t|s_t)\right)\right]
\end{aligned}
$$

Taking the derivative w.r.t. $\theta$,

$$\frac{\partial}{\partial \theta}\mathcal{L}(\theta) = \sum_{t=0}^{T} E_\mathcal{D}\left[\frac{\partial}{\partial \theta}f_\theta(s_t, a_t)\right] - E_{\mu_t}\left[\frac{\exp\{f_\theta(s_t, a_t)\}}{\frac{1}{2}\exp\{f_\theta(s_t, a_t)\} + \frac{1}{2}\pi(a_t|s_t)}\frac{\partial}{\partial \theta}f_\theta(s_t, a_t)\right]$$

Multiplying the top and bottom of the fraction in the second expectation by the state marginal $\pi(s_t) = \int_a \pi_t(s_t, a_t)$, and grouping terms we get:

$$\frac{\partial}{\partial \theta}\mathcal{L}(\theta) = \sum_{t=0}^{T} E_\mathcal{D}\left[\frac{\partial}{\partial \theta}f_\theta(s_t, a_t)\right] - E_\mu\left[\frac{\hat{p}_{\theta,t}(s_t, a_t)}{\hat{\mu}_t(s_t, a_t)}\frac{\partial}{\partial \theta}f_\theta(s_t, a_t)\right]$$

Where we have written $\hat{p}_{\theta,t}(s_t, a_t) = \exp\{f_\theta(s_t, a_t)\}\pi_t(s_t)$, and $\hat{\mu}$ to denote a mixture between $\hat{p}_\theta(s, a)$ and policy samples.

This expression matches Eqn. 5, with $f_\theta(s, a)$ serving as the reward function, when $\pi$ maximizes the policy objective so that $\hat{p}_\theta(s, a) = p_\theta(s, a)$.

## A.3 POLICY OBJECTIVE

Next, we show that the policy objective matches that of the sampler of Eqn. 6. The objective of the policy is to maximize with respect to the reward $\hat{r}_t(s, a)$. First, note that:

$$
\begin{aligned}
\hat{r}_t(s, a) &= \log(D_\theta(s, a)) - \log(1 - D_\theta(s, a)) \\
&= \log \frac{e^{f_\theta(s, a)}}{e^{f_\theta(s, a)} + \pi(a|s)} - \log \frac{\pi(a|s)}{e^{f_\theta(s, a)} + \pi(a|s)} \\
&= f_\theta(s, a) - \log \pi(a|s)
\end{aligned}
$$

Thus, when $\hat{r}(s, a)$ is summed over entire trajectories, we obtain the entropy-regularized policy objective

$$E_\pi\left[\sum_{t=0}^{T}\hat{r}_t(s_t, a_t)\right] = E_\pi\left[\sum_{t=0}^{T}f_\theta(s_t, a_t) - \log \pi(a_t|s_t)\right]$$

Where $f_\theta$ serves as the reward function.

## A.4 $f_\theta(s, a)$ RECOVERS THE ADVANTAGE

The global minimum of the discriminator objective is achieved when $\pi = \pi_E$, where $\pi$ denotes the learned policy (the "generator" of a GAN) and $\pi_E$ denotes the policy under which demonstrations were collected (Goodfellow et al., 2014). At this point, the output of the discriminator is $\frac{1}{2}$ for all values of $s, a$, meaning we have $\exp\{f_\theta(s, a)\} = \pi_E(a|s)$, or $f^*(s, a) = \log \pi_E(a|s) = A^*(s, a)$.

# B  STATE-ONLY INVERSE REINFORCEMENT LEARNING

In this section we include proofs for Theorems 5.1 and 5.2, and the condition on the dynamics necessary for them to hold.

**Definition B.1** (Decomposability Condition). *Two states $s_1, s_2$ are defined as "1-step linked" under a dynamics or transition distribution $T(s'|a, s)$ if there exists a state $s$ that can reach $s_1$ and $s_2$ with positive probability in one time step. Also, we define that this relationship can transfer through transitivity: if $s_1$ and $s_2$ are linked, and $s_2$ and $s_3$ are linked, then we also consider $s_1$ and $s_3$ to be linked.*

*A transition distribution $T$ satisfies the decomposability condition if all states in the MDP are linked with all other states.*

The key reason for needing this condition is that it allows us to decompose the functions state dependent $f(s)$ and next state dependent $g(s')$ from their sum $f(s) + g(s')$, as stated below:

**Lemma B.1.** *Suppose the dynamics for an MDP satisfy the decomposability condition. Then, for functions $a(s), b(s), c(s), d(s)$, if for all $s, s'$:*

$$a(s) + b(s') = c(s) + d(s')$$

*Then for for all $s$,*

$$a(s) = c(s) + const$$
$$b(s) = d(s) + const$$

*Proof.* Rearranging, we have:

$$a(s) - c(s) = b(s') - d(s')$$

Let us rewrite $f(s) = a(s) - c(s)$. This means we have $f(s) = b(s') - d(s')$ for some function only dependent on $s$. In order for this to be representable, the term $b(s') - d(s')$ must be equal for all successor states $s'$ from $s$. Under the decomposability condition, all successor states must therefore be equal in this manner through transitivity, meaning we have $b(s') - d(s')$ must be constant with respect to $s$. Therefore, $a(s) = c(s) + \text{const}$. We can then substitute this expression back in to the original equation to derive $b(s) = d(s) + \text{const}$. $\square$

We consider the case when the ground truth reward is state-only. We now show that if the learned reward is also state-only, then we guarantee learning disentangled rewards, and vice-versa (sufficiency and necessity).

**Theorem 5.1.** *Let $r(s)$ be a ground-truth reward, and $T$ be a dynamics model satisfying the decomposability condition. Suppose IRL recovers a state-only reward $r'(s)$ such that it produces an optimal policy in $T$:*

$$Q^*_{r',T}(s, a) = Q^*_{r,T}(s, a) - f(s)$$

*Then, $r'(s)$ is disentangled with respect to all dynamics.*

*Proof.* We show that $r'(s)$ must equal the ground-truth reward up to constants (modifying rewards by constants does not change the optimal policy).

Let $r'(s) = r(s) + \phi(s)$ for some arbitrary function of state $\phi(s)$. We have:

$$Q^*_r(s, a) = r(s) + \gamma E_{s'}[\operatorname*{softmax}_{a'} Q^*_r(s', a')]$$

$$Q^*_r(s, a) - f(s) = r(s) - f(s) + \gamma E_{s'}[\operatorname*{softmax}_{a'} Q^*_r(s', a')]$$

$$Q^*_r(s, a) - f(s) = r(s) + \gamma E_{s'}[f(s')] - f(s) + \gamma E_{s'}[\operatorname*{softmax}_{a'} Q^*_r(s', a') - f(s')]$$

$$Q^*_{r'}(s, a) = r(s) + \gamma E_{s'}[f(s')] - f(s) + \gamma E_{s'}[\operatorname*{softmax}_{a'} Q^*_{r'}(s', a')]$$

From here, we see that:

$$r'(s) = r(s) + \gamma E_{s'}[f(s')] - f(s)$$

Meaning we must have for all $s, a$:

$$\phi(s) = \gamma E_{s'}[f(s')] - f(s)$$

This places the requirement that all successor states from $s$ must have the same potential $f(s)$. Under the decomposability condition, every state in the MDP can be linked with such an equality statement, meaning that $f(s)$ is constant. Thus, $r'(s) = r(s) + \text{const}$. $\square$

**Theorem 5.2.** *If a reward function $r'(s, a, s')$ is disentangled for all dynamics functions, then it must be state-only. i.e. If for all dynamics $T$,*

$$Q^*_{r,T}(s, a) = Q^*_{r',T}(s, a) + f(s) \; \forall s, a$$

*Then $r'$ is only a function of state.*

*Proof.* We show the converse, namely that if $r'(s, a, s')$ can depend on $a$ or $s'$, then there exists a dynamics model $T$ such that the optimal policy is changed, i.e. $Q^*_{r,T}(s, a) \neq Q^*_{r',T}(s, a) + f(s) \; \forall s, a$.

Consider the following 3-state MDP with deterministic dynamics and starting state $S$:

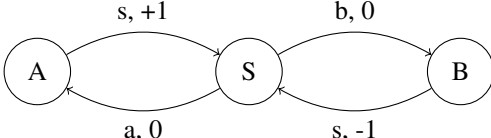

We denote the action with a small letter, i.e. taking the action $a$ from $S$ brings the agent to state $A$, receiving a reward of $0$. For simplicity, assume the discount factor $\gamma = 1$. The optimal policy here takes the $a$ action, returns to $s$, and repeat for infinite positive reward. An action-dependent reward which induces the same optimal policy would be to move the reward from the action returning to $s$ to the action going to $a$ or $s$:

$$r'(s, a) = \quad \begin{array}{|c|c|c|} \hline \text{State} & \text{Action} & \text{Reward} \\ \hline S & a & +1 \\ S & b & -1 \\ A & s & 0 \\ B & s & 0 \\ \hline \end{array}$$

This corresponds to the shaping potential $\phi(S) = 0, \phi(A) = 1, \phi(B) = -1$.

Now suppose we modify the dynamics such that action $a$ leads to $B$ and action $b$ leads to $A$:

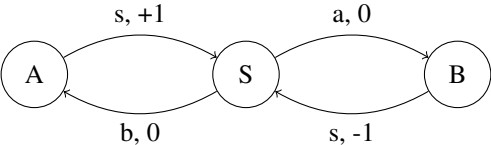

Optimizing $r'$ on this new MDP results in a different policy than optimizing $r$, as the agent visits B, resulting in infinite negative reward. $\square$

## C   AIRL RECOVERS REWARDS UP TO CONSTANTS

In this section, we prove that AIRL can recover the ground truth reward up to constants if the ground truth is only a function of state $r(s)$. For simplicity, we consider deterministic environments, so that $s'$ is uniquely defined by $s, a$, and we restrict AIRL's reward estimator $g$ to only be a function of state.

**Theorem C.1.** *Suppose we use AIRL with a discriminator of the form*

$$f(s, a, s') = g_\theta(s) + \gamma h_\phi(s') - h_\phi(s)$$

*We also assume we have deterministic dynamics, in addition to the decomposability condition on the dynamics from Thm 5.1.*

*Then if AIRL recovers the optimal $f^*(s, a, s')$, we have*

$$g_\theta^*(s) = r(s) + const$$

$$h_\phi^*(s) = V^*(s) + const$$

*Proof.* From Appendix A.4, we have $f^*(s, a, s') = A^*(s, a)$, so $f^*(s, a, s') = Q^*(s, a) - V^*(s) = r(s) + \gamma V^*(s') - V^*(s)$.

Substituting the form of $f$, we have for all $s, s'$:

$$g^*(s) + \gamma h^*(s') - h^*(s) = r(s) + \gamma V^*(s') - V^*(s)$$

Applying Lemma B.1 with $a(s) = g^*(s) - h^*(s)$, $b(s') = \gamma h^*(s')$, $c(s) = r(s) - V^*(s)$, and $d(s') = \gamma V^*(s')$ we have the result. $\qquad\square$

# D  EXPERIMENT DETAILS

In this section we detail hyperparameters and training procedures used for our experiments. These hyperparameters were selected via a grid search.

## D.1  NETWORK ARCHITECTURES

For the tabular MDP environment, we also use a tabular representation for function approximators.

For continuous control experiments, we use a two-layer ReLU network with 32 units for the discriminator of GAIL and GAN-GCL. For AIRL, we use a linear function approximator for the reward term $g$ and a 2-layer ReLU network for the shaping term $h$. For the policy, we use a two-layer (32 units) ReLU gaussian policy.

## D.2  OTHER HYPERPARAMETERS

*Entropy regularization*: We use an entropy regularizer weight of $0.1$ for Ant, Swimmer, and HalfCheetah across all methods. We use an entropy regularizer weight of $1.0$ on the point mass environment.

*TRPO Batch Size*: For Ant, Swimmer and HalfCheetah environments, we use a batch size of 10000 steps per TRPO update. For pendulum, we use a batch size of 2000.

## D.3  OTHER TRAINING DETAILS

IRL methods commonly learn rewards which explain behavior locally for the current policy, because the reward can "forget" the signal that it gave to an earlier policy. This makes rewards obtained at the end of training difficult to optimize from scratch, as they overfit to samples from the current iteration. To somewhat migitate this effect, we mix policy samples from the previous 20 iterations of training as negatives when training the discriminator. We use this strategy for both AIRL and GAN-GCL.

