# OpenReview forum: "Learning Robust Rewards with Adverserial Inverse Reinforcement Learning"
_ICLR.cc/2018/Conference — Accept (Poster)_

### Official Review · AnonReviewer3 · 2017-11-26
**Learns environment-independent rewards; reasonable next step in adversarial IRL**

**Rating:** 6
**Confidence:** 4

**Review:**

The paper provides an approach to learning reward functions in high-dimensional domains, showing that it performs comparably to other recent approaches to this problem in the imitation-learning setting. It also argues that a key property to learning generalizable reward functions is for them to depend on state, but not state-action or state-action-state. It uses this property to produce "disentangled rewards", demonstrating that they transfer well to the same task under different transition dynamics.

The need for "state-only" rewards is a useful insight and is covered fairly well in the paper. The need for an "adversarial" approach is not justified as fully, but perhaps is a consequence of recent work. The experiments are thorough, although the connection to the motivation in the abstract (wanting to avoid reward engineering) is weak.

Detailed feedback:

"deployed in at test-time on environments" -> "deployed at test time in environments"?

"which can effectively recover disentangle the goals" -> "which can effectively disentangle the goals"?

"it allows for sub-optimality in demonstrations, and removes ambiguity between demonstrations and the expert policy": I am not certain what is being described here and it doesn't appear to come up again in the paper. Perhaps remove it?

"r high-dimensional (Finn et al., 2016b) Wulfmeier" -> "r high-dimensional (Finn et al., 2016b). Wulfmeier".

"also consider learning cost function with" -> "also consider learning cost functions with"?

"o learn nonlinear cost function have" -> "o learn nonlinear cost functions have".

" are not robust the environment changes" -> " are not robust to environment changes"?

"We present a short proof sketch": It is unclear to me what is being proven here. Please state the theorem.

"In the method presented in Section 4, we cannot learn a state-only reward function": I'm not seeing that. Or, maybe I'm confused between rewards depending on s vs. s,a vs. s,a,s'. Again, an explicit theorem statement might remove some confusion here.

"AIRLperforms" -> "AIRL performs".

Figure 2: The blue and green colors look very similar to me. I'd recommend reordering the legend to match the order of the lines (random on the bottom) to make it easier to interpret.

"must reach to goal" -> "must reach the goal"?

"pointmass" -> "point mass". (Multiple times.)

Amin, Jiang, and Singh's work on efficiently learning a transferable reward function seems relevant here. (Although, it might not be published yet: https://arxiv.org/pdf/1705.05427.pdf.)

Perhaps the final experiment should have included state-only runs. I'm guessing that they didn't work out too well, but it would still be good to know how they compare.

---

> ### Author Response · Authors · 2018-01-05
> **Response to AnonReviewer3**
>
> Thank you for the detailed feedback. We have included all of the typo corrections and clarifications, as well as included state-only runs in the imitation learning experiments (Section 7.3). As detailed below, we believe that we have addressed all of the issues raised in your review, but we would appreciate any further feedback you might offer.
>
> > The need for an "adversarial" approach is not justified as fully, but perhaps is a consequence of recent work.
>
> Adversarial approaches are an inherent consequence of using sampling-based methods for training energy-based models, and we’ve edited Section 2, paragraph 2 to make this more clear. There is in fact no other (known) choice for doing this: any method that does maxent IRL and generates samples (rather than assuming known dynamics) must be adversarial in nature, as shown by Finn16a. Traditional methods like tabular MaxEnt IRL [Ziebart 08] have an adversarial nature as they must alternate between an inner-loop RL problem (the sampler) and updating the reward function (the discriminator).
>
> > Although the connection to the motivation in the abstract (wanting to avoid reward engineering) is weak.
>
> We’ve slightly modified the paragraph before section 7.1 to make this connection more clear. We use environments where a reward function is available for the purpose of easily collecting demonstrations (otherwise we would need to resort to motion capture or teleoperation). However the experimental setup after demo collection is exactly the same as one would encounter while using IRL when a ground truth reward is not available.
>
> > Amin, Jiang, and Singh's work on efficiently learning a transferable reward function seems relevant here. (Although, it might not be published yet: https://arxiv.org/pdf/1705.05427.pdf.)
>
> Amin, Jian & Singh’s work is indeed relevant and we have also included it in the related work section.
>
> > Perhaps the final experiment should have included state-only runs. I'm guessing that they didn't work out too well, but it would still be good to know how they compare.
>
> We’ve included these in the experiments. State-only runs perform slightly worse as expected, since the true reward has torque penalty terms which depend on the action, and cannot be captured by the model. However the performance isn’t so bad that the agent fails to solve the task.

---

### Official Review · AnonReviewer2 · 2017-11-28
**Using the deterministic-MDP formulation of MaxEnt IRL is a concern**

**Rating:** 6
**Confidence:** 2

**Review:**

SUMMARY:
This paper considers the Inverse Reinforcement Learning (IRL) problem, and particularly suggests a method that obtains a reward function that is robust to the change of dynamics of the MDP.

It starts from formulating the problem within the MaxEnt IRL framework of Ziebart et al. (2008). The challenge of MaxEnt IRL is the computation of a partition function. Guided Cost Learning (GCL) of Finn et al. (2016b) is an approximation of MaxEnt IRL that uses an adaptive importance sampler to estimate the partition function. This can be shown to be a form of GAN, obtained by using a specific discriminator [Finn et al. (2016a)].

If the discriminator directly works with trajectories tau, the result would be GAN-GCL. But this leads to high variance estimates, so the paper suggests using a single state-action formulation, in which the discriminator f_theta(s,a) is a function of (s,a) instead of the trajectory. The optimal solution of this discriminator is to have f(s,a) = A(s,a) — the advantage function.
The paper, however, argues that the advantage function is “entangled” with the dynamics, and this is undesirable. So it modified the discriminator to learn a function that is a combination of two terms, one only depends on state-action and the other depends on state, and has the form of shaped reward transformation.


EVALUATION:

This is an interesting paper with good empirical results. As I am not very familiar with the work of Finn et al. (2016a) and Finn et al. (2016b), I have not verified the detail of derivations of this new paper very closely. That being said, I have some comments and questions:


* The MaxEnt IRL formulation of this work, which assumes that p_theta(tau) is proportional to exp( r_theta (tau) ), comes from
[Ziebart et al., 2008] and assumes a deterministic dynamics. Ziebart’s PhD dissertation [Ziebart, 2010] or the following paper show that the formulation is different for stochastic dynamics:

Ziebart, Bagnell, Dey, “The Principle of Maximum Causal Entropy for Estimating Interacting Processes,” IEEE Trans. on IT, 2013.

Is it still a reasonable thing to develop based on this earlier, an inaccurate, formulation?


* I am not convinced about the argument of Appendix C that shows that AIRL recovers reward up to constants.
It is suggested that since the only items on both sides of the equation on top of p. 13 depend on s’ are h* and V, they should be equal.
This would be true if s’ could be chosen arbitrararily. But s’ would be uniquely determined by s for a deterministic dynamics. In that case, this conclusion is not obvious anymore.

Consider the state space to be integers 0, 1, 2, 3, … .
Suppose the dynamics is that whenever we are at state s (which is an integer), at the next time step the state decreases toward 1, that is s’ = phi(s,a) = s - 1; unless s = 0, which we just stay at s’ = s = 0. This is independent of actions.
Also define r(s) = 1/s for s>=1 and r(0) = 0.
Suppose the discount factor is gamma = 1 (note that in Appendix B.1, the undiscounted case is studied, so I assume gamma = 1 is acceptable).

With this choices, the value function V(s) = 1/s + 1/(s-1) + … + 1/1 = H_s, i.e., the Harmonic function.
The advantage function is zero. So we can choose g*(s) = 0, and h*(s) = h*(s’) = 1.
This is in contrast to the conclusion that h*(s’) = V(s’) + c, which would be H_s + c, and g*(s) = r(s) = 1/s.
(In fact, nothing is special about this choice of reward and dynamics.)

Am I missing something obvious here?

Also please discuss how ergodicity leads to the conclusion that spaces of s’ and s are identical. What does “space of s” mean? Do you mean the support of s? Please make the argument more rigorous.


* Please make the argument of Section 5.1 more rigorous.

---

> ### Public Comment · (anonymous) · 2017-12-03
> **Some discussion for this review**
>
> Yes, you missed an important fact for f*(s,a) = log_\pi_E(a|s) = A*(s,a), this equality holds only under the policy are updated with a max entropy regularization(you can refer to this article: http://arxiv.org/abs/1702.08165). And under this context, Q and V are not just the the reward sum but with an extra entropy item. So it may be not correct to use such a simple example. In addition, since the methods in this paper are all with the max entropy item, it is ok for the authors to use this form of result.

---

> ### Author Response · Authors · 2018-01-05
> **Response to AnonReviewer2**
>
> Thank you for the constructive feedback. We’ve incorporated your comments and clarified certain points of the paper below. Please let us know if there are other additional issues which need clarification.
>
> > The MaxEnt IRL formulation of this work, which assumes that p_theta(tau) is proportional to exp( r_theta (tau) ), comes from
> [Ziebart et al., 2008] and assumes a deterministic dynamics. Ziebart’s PhD dissertation [Ziebart, 2010] or the following paper show that the formulation is different for stochastic dynamics.
> Is it still a reasonable thing to develop based on this earlier, an inaccurate, formulation?
>
> We have updated the background (section 3) and appendix (section A) to use the maximum causal entropy framework rather than the earlier maximum entropy framework of [Ziebart 08]. Our algorithm requires no changes since the causal entropy framework more accurately describes what we were doing in the first place (our old derivations were valid in the deterministic case, where MaxEnt and MaxCausalEnt are identical, but in the stochastic case, our approach in fact matches MaxCausalEnt).
>
> > * I am not convinced about the argument of Appendix C that shows that AIRL recovers reward up to constants.
> Also please discuss how ergodicity leads to the conclusion that spaces of s’ and s are identical. What does “space of s” mean? Do you mean the support of s? Please make the argument more rigorous.
> * Please make the argument of Section 5.1 more rigorous.
>
> We’ve provided more formal proofs for Section 5 and the appendix. In order to fix the statements, we’ve changed the condition on the dynamics - a major component is that it requires that each state be reachable from >1 other state within one step. Ergodicity is neither a sufficient nor necessary condition on the dynamics, but special cases such as an ergodic MDP with self-transitions at each state satisfies the new condition (though the minimum necessary conditions are less restrictive).

---

> > ### Comment · AnonReviewer2 · 2018-01-13
> > **After rebuttal**
> >
> > Thank you for your reply and the revision of the paper. I briefly gone through the revised paper. My concerns have been addressed (but I should say that I have not verified the math closely).

---

### Official Review · AnonReviewer1 · 2017-12-01
**A variante of the GAN-GCL for Inverse RL (IRL) is presented and evaluated. The difference with the original algorithm is the fact that the sampling happens at the level of stat-actions instead of full trajectories, to reduce its variance. Empirical results clearly show the advantage of this method.**

**Rating:** 7
**Confidence:** 3

**Review:**

This paper revisits the generative adversarial network guided cost learning (GAN-GCL)  algorithm presented last year. The authors argue learning rewards from sampled trajectories has a high variance. Instead, they propose to learn a generative model wherein actions are sampled as a function of states. The same energy model is used for sampling actions: the probability of an action is proportional to the exponential of its reward. To avoid overfitting the expert's demonstrations (by mimicking the actions directly instead of learning a reward that can be generalized to different dynamics), the authors propose to learn rewards that depend only on states, and not on actions. Also, the proposed reward function includes a shaping term, in order to cover all possible transformations of the reward function that could have been behind the expert's actions. The authors argue formally that this is necessary to disentangle the reward function from the dynamics. Th paper also demonstrates this argument empirically (e.g. Figure 1).

This paper is well-written and technically sound. The empirical evaluations seem to be supporting the main claims of the paper. The paper lacks a little bit in novelty since it is basically a variante of GAN-GCL, but it makes it up with the inclusion of  a shaping term in the rewards and with the related formal arguments. The empirical evaluations could also be strengthened with experiments in higher-dimensional systems (like video games).

"Under maximum entropy IRL, we assume the demonstrations are drawn from an optimal policy p(\tau) \propto exp(r(tau))" This is not an assumption, it's the form of the solution we get by maximizing the entropy (for regularization).

---

> ### Public Comment · (anonymous) · 2017-12-03
> **Some discussion for the last sentence of this review**
>
> I think "Under maximum entropy IRL, we assume the demonstrations are drawn from an optimal policy p(\tau) \propto exp(r(tau))" is right. Since we sampled from expert policy or just made some demonstrations, and we were unaware to this max entropy items when we did this, we can only assume that these trajs obeying the boltzman distribution. In fact, in the context of MaxEnt IRL, only the optimal trajs distribution output by the model(by maximizing the max entropy item under the feature expectaton constraints) has a closed form of \propto exp(r(tau)).

---

> ### Author Response · Authors · 2018-01-05
> **Response to AnonReviewer1**
>
> Thank you for the thoughtful feedback. We’ve incorporated the suggestions to the best of our ability, and clarified portions of the paper, as described below.
>
> > "Under maximum entropy IRL, we assume the demonstrations are drawn from an optimal policy p(\tau) \propto exp(r(tau))" This is not an assumption, it's the form of the solution we get by maximizing the entropy (for regularization).
>
> We’ve modified Section 3 to remove this ambiguity (note that we’ve also modified the section to use the causal entropy framework as requested by another reviewer). This statement was referring to the fact that we are assuming the expert is drawing samples from the distribution p(tau), not the fact that p(tau) \propto exp(r(tau)).
>
> > "The paper lacks a little bit in novelty since it is basically a variant of GAN-GCL, but it makes it up with the inclusion of  a shaping term in the rewards and with the related formal arguments."
>
> In regard to GAN-GCL, we would note that, although the method draws heavily on the theory in this workshop paper, it is unpublished and does not describe an implementation of any actual algorithm -- the GAN-GCL paper simply describes a theoretical connection between GANs and IRL. Our implementation of the algorithm that is closest to the one suggested by the theory in the GAN-GCL workshop paper does not perform very well in practice (Section 7.3).

---

### Public Comment · ~Max_Morrison1 · 2017-11-13
**ICLR 2018 Reproducibility Workshop Inquiry**

Hello authors,

We are three students at the University of Michigan working together to submit to the ICLR 2018 Reproducibility Workshop.  We are all very interested in the AIRL algorithm described in your paper “Learning Robust Rewards with Adversarial Inverse Reinforcement Learning” and would like to focus on it for our submission.  Specifically, we hope to both reproduce your results as well as conduct further experiments and hyperparameter tuning.  Our team would greatly benefit from working with your implementation.  Your paper mentioned that you intend to post the implementation of AIRL.  Do you happen to have a schedule for when the code will be available for us to use?

Thank you for your time and your work,
Jin, Max, and Sam

---

> ### Author Response · Authors · 2017-11-13
> **Code is partially released (will fully release after acceptance)**
>
> Hi Jin, Max, and Sam
>
> We won't release the full code until after acceptance, but we have already released a publicly available implementation of the baselines + the "non-robust" version of AIRL. This should be a very good starting point for reproducibility. If you can provide an email address, we can send you a link (so as to not break anonymity on OpenReview).

---

> > ### Public Comment · ~Max_Morrison1 · 2017-11-13
> > **Contact Info**
> >
> > Hello authors,
> >
> > That would be an excellent starting point. Our email addresses are as follows:
> >
> >   - Jin (wyjin@umich.edu)
> >   - Max (morrimax@umich.edu)
> >   - Sam (samtenka@umich.edu)
> >
> > Thank you for your help,
> > Jin, Max, and Sam

---

### Public Comment · ~David_Anthony_Venuto1 · 2017-11-25
**Code Release for ICLR Reproducibility Workshop**

Hello,

We are also a group of 3 students from McGill University, who are participating in the ICLR 2018 Reproducibility Research.  We are also interested in the AIRL algorithm proposed in your paper and validating the results found in your paper. We noticed that you released a portion of your code, namely the non-robust version of your AIRL algorithm to other participants in this reproducibility challenge. We were wondering if you could also provide us with that link as it will go a long way to assisting us in reproducing your results.

For your convenience, we have provided our emails below.

Best Regards,

Isaac (isaac.chan@mail.mcgill.ca)
David (david.venuto@mail.mcgill.ca)

---

### Public Comment · (anonymous) · 2017-12-16
**ICLR 2018 Reproducibility Review**

We reproduce the results from the submitted ICLR paper: "Learning Robust Rewards with Adversarial Inverse Reinforcement Learning" where we reproduce the previous state of the art results, namely the Generative Adversarial Network - Guided Cost Learning (Finn et. al 2016), the Generative Adversarial Imitation Learning (Ho & Ermon 2016) and make a comparison to the non-robust version of the AIRL algorithm (a previous iteration of the robust version of the AIRL algorithm) methods on the pendulum, custom designed ant and pointmass openAI gym environments.

This paper introduces an inverse reinforcement learning technique that utilizes a Generative Adversarial Network (GAN) (Goodfellow et. al 2014) to generate its rewards called Adversarial Inverse Reinforcement Learning (AIRL) (Anonymous, 2018). The algorithm updates the discriminator by training with expert trajectories, and then updating the policy in an attempt to confuse the discriminator. The paper makes two main claims about the functionality of the algorithm: 1) AIRL can learn robust rewards that make it the optimal algorithm for transfer learning tasks. 2) That AIRL is scalable up to high-dimensional tasks. This paper goes on to further claim that AIRL can perform competitively in imitation learning environments when compared to previous state-of-the-art algorithms such as generative adversarial imitation learning (GAIL) (Ho & Ermon 2016), but when it comes to transfer learning tasks, AIRL performs significantly better compared to those same algorithms.

While we could not implement the robust AIRL algorithm, we made the effort to do as many experiments with the baseline algorithms. We believe that our inability to reproduce the full robust AIRL algorithm is not a statement on the reproducibility of this paper, but simply due to our lack of technical expertise. The results of these methods are as follows:

Transfer Learning Experiments:

        Method & Pointmass-Maze & Disabled Ant
        GAN-GCL & -61.8  & -79.201
        AIRL & -51.2  & -92.578
        GAIL & -40.2  & -70.5668
        TRPO (Expert Policy) & -17.1 & 150.7

Imitation Learning Experiments:

        Method & Pendulum & Ant
        GAN-GCL & -242.5 & 467.7
        AIRL & -210.7 & 983.7
        GAIL & -198.2 & 1501.3
        TRPO (Expert Policy) & -128.4 & 2000.6

As can be seen from these tables, our results and those found in the paper seem to be fairly similar. Our lower results with the AIRL algorithm is to be expected as we implemented the non-robust version whereas the paper shows results for the robust version. The variance in our results could be due to the unspecified n_iteration parameter, where higher/lower values of the n_iteration could contribute to higher/lower scores respectively.

Our choice of hyperparameters were effectively those found in the report and the default hyperparameters found in the code provided to us by the authors. We increased the number of iterations to 1500 for the ant and pointmaze tasks, and increased it to 500 for the pendulum task to increase the chance of convergence.

Our choice of selecting these environments was to test the claims that AIRL can not only be effective in transfer learning tasks, but to also scale up to high dimensional environments. Our empirical observations suggested that the ant/disabled ant task provided the highest dimensional environment for which we could test the scalability. Running on 7th Generation Intel® Core™ i7-7700HQ Quad Core Processor at 2.6GHz took 1 hour and 30 minutes to run 1000 iterations of the non-robust AIRL algorithm on the disabled ant task.

As far as the level of reproducibility is concerned. Having been provided the code from the authors went a long way to helping us reproduce the experiment. Within the code, the custom environments and the baseline algorithms were all provided which helped ensure that we were conducting the experiments in a fairly similar environment. Even though we could not reproduce the robust version of the AIRL algorithm, the mathematical foundations for the algorithm, along with pseudocode of the algorithm itself, is stated very clearly in the paper. As a result, our conjecture is that anyone who is more experienced in implementing code in RLLAB and openAIgym should have relatively little difficulty in implementing the robust version of the AIRL algorithm, but we do not have the expertise to state this for fact.

---

### Public Comment · (anonymous) · 2017-12-21
**What if the groundtruth reward is a function of both the state and the action?**

In figure 1, the authors show an example of state only reward recovers the groundtruth. However, the groundtruth reward here is a function of only state. What if the groundtruth reward is a function of both the state and the action, can we still apply this method?

In the continuous control experiment 2, if the ant achieves reward by moving forward. It is obvious that the reward depends on both s and s', I'm confused how a reward solely depending on s is able to recover the groundtruth.

---

> ### Author Response · Authors · 2018-01-19
> **No performance guarantees, but may work well in practice depending on the environment**
>
> If the ground truth reward depends on both states and actions, the algorithm cannot represent the true reward and thus the performance of the policy will not match that of the experts (we have included new experiments in Section 7.3 for this case). The results will likely depend on the task - in our experiments the performance was not much worse than the experts, but the only action-dependent term in the reward for OpenAI Gym locomotion tasks is a control penalty for actions with large magnitude.
>
> However, we also argue that no IRL algorithm which operates over arbitrary reward function classes will be able to recover ground truth rewards in this case, since we cannot avoid reward shaping (section 5). In order to remove shaping, we need to manually restrict the class of reward functions such that shaping is not possible. An alternative approach is to adopt a multi-task IRL paradigm to generalize across different dynamics.
>
> The state definition for most OpenAI Gym locomotion tasks (including the ones used in this paper) contains velocities - thus we can still represent the ground truth reward.

---

### Public Comment · ~Xiaojian_Ma1 · 2018-01-17
**Where is the partition function Z?**

Under the assumption of MaxEnt IRL, demonstration can be seen as trajectories drawn from (1/Z)exp(-c(\tau)), however, in eq(2) the partition function disappears(compared to Sec 3.1 in http://arxiv.org/abs/1611.03852).  Correct me if I am wrong,  as the authors propose f*(\tau) = R*(\tau) + const(R is an entropy regularized reward, that's ok), do they mean by the logZ item(if the partition function exists in eq(2)) can be a constant?  And refer to Appendix A.4, f* == log\pi_E == A*(s,a), the second part of this equation comes from soft Q-learning because of the entropy regularized reward R, the the first part holds because it eliminates the partition function Z, so I wonder if it still holds even when the partition function is added?

And there is another minor typo(?) The inline equation under eq(2): R(\tau) = log(1-D(\tau)) - log(D(\tau)) -> R(\tau) = log(D(\tau)) - log(1-D(\tau)), according to Appendix A.3

---

> ### Author Response · Authors · 2018-01-19
> **The partition function is learned as part of f(\tau)**
>
> Section 3.1 of Finn 2016 (http://arxiv.org/abs/1611.03852) is incorrect in regard to learning the partition function on the bias of the last sigmoid layer. We can't uniquely separate the bias term from the rest of the function approximator. For example, the cost function approximator c_\theta(tau) could incorporate the log Z term and we could set the learned bias term to 0. Thus, there is no point in explicitly adding a separate learned bias term to capture the partition function as in Finn16 - we simply learn a function f(\tau) which implicitly learns the partition function, although we cannot extract it.

---

### Public Comment · (anonymous) · 2019-03-12
**About the GAIL baseline**

I appreciate your insightful paper on learning a disentangled reward via adversarial training. However, I have a question in your experiment settings, and I hope you can help me with that.

In Table.1/2, what are the differences between AIRL w/o state only(State-only is 'No') and GAIL? As stated at the beginning of Section 4, it seems that AIRL with (s,a) as the input of discriminator (in my understanding, this is just AIRL with State-only is 'No', correct me if I'm wrong) has almost the same objective as GAIL (see Appendix A.3).

I suppose the difference comes from the choice of entropy regularizer weight since GAIL set it to 0 for all the tasks other than reacher. But I'm not sure as it mentioned at the end of section 3 that GAIL does not place a special structure on the discriminator, and an entropy regularizer is obviously not such a special structure.

---

### Decision · Program_Chairs · 2018-01-29
**ICLR 2018 Conference Acceptance Decision**

**Decision:**

Accept (Poster)

**Comment:**

The AIRL is presented as a scalable inverse reinforcement learning algorithm. A key idea is to produce "disentangled rewards", which are invariant to changing dynamics; this is done by having the rewards depend only on the current state. There are some similarities with GAIL and the authors argue that this is effectively a concrete implementation of GAN-GCL that actually works.  The results look promising to me and the portability aspect is neat and useful!

In general, the reviewers found this paper and its results interesting and I think the rebuttal addressed many of the concerns. I am happy that the reproducibility report is positive which helped me put this otherwise potentially borderline paper into the 'accept' bucket.